# Implementation of the Robson Classification in Greece: A Retrospective Cross-Sectional Study

**DOI:** 10.3390/healthcare11060908

**Published:** 2023-03-21

**Authors:** Paraskevi Giaxi, Kleanthi Gourounti, Victoria Vivilaki, Panagiotis Zdanis, Antonis Galanos, Aris Antsaklis, Aikaterini Lykeridou

**Affiliations:** 1Department of Midwifery, University of West Attica, 12243 Athens, Greece; 2Laboratory for Research of the Musculoskeletal System, School of Medicine, National and Kapodistrian University of Athens, 15784 Athens, Greece; 3IASO, General Maternity and Gynecology Clinic, 15123 Athens, Greece

**Keywords:** audit, cesarean section, rate, Greece, Robson classification

## Abstract

Cesarean sections have become the most commonly performed operations around the world. The World Health Organization recommended the use of the Robson classification system as a universal standard to establish a joint control system in healthcare facilities. The aim of this study was to implement the Robson classification for the first time in Greece to identify trends in cesarean births and examine the groups of women who are the main contributors to the increasing rates. Moreover, the indicators for cesarean sections will be evaluated as per the Robson classification. In the sample analysis, we included the records of 8572 women giving birth in one private health facility in Greece. A total of 8572 women gave birth during the study period, of which 5224 (60.9%) were cesarean section births and 3348 (39.1%) were vaginal births. In our study, according to the Robson classification, the largest contributors to the overall CS rate were as follows: (a) nulliparous women with a single cephalic term pregnancy, who were either labor induced or delivered by cesarean section before labor—Group 2 (34.6%); (b) multiparous women with a single cephalic term pregnancy and at least one previous cesarean section—Group 5 (30.7%); (c) women with a single cephalic preterm pregnancy—Group 10 (11.7%); (d) women with multiple pregnancies—Group 8 (7.0%). Our study is expected to assist policymakers in Greece in planning further interventions for each subgroup of women in order to reduce the overall CS rate and unnecessary CSs.

## 1. Introduction

The cesarean section has become the most commonly performed operation around the world, following an uneven pattern [1]. On the one hand, the insufficient use of cesarean sections based on indications, which is mainly seen in developing countries, increases perinatal morbidity and mortality; on the other hand, the abuse or unnecessary use of cesarean sections (e.g., increased percentage of pregnancy terminations by cesarean sections without medical or obstetric indications), which is mainly seen in developed countries, does not seem to offer advantages but, on the contrary, may cause implications to the mother and/or the infant, while, at the same time, it increases the cost of financial resources (e.g., human resources, increased hospitalization, etc.) [2,3,4]. Worldwide, from 1990 to 2018, the percentage of cesarean sections has increased by 19%, while according to international research opinions, it is expected to increase from 21.1% in 2018 to 28.5% by 2030 [5]. The statement of the World Health Organization in 1985 that the ideal percentage of cesarean sections is between 10% and 15% was based on the scientific data available at the time; however, today the validity of this margin has been questioned by the world scientific community [6]. There is no evidence on what the optimal mode of birth is. Likewise, there are variations in the CS rates amongst institutions, regions, and countries due to many factors, such as variations in patient characteristics or preferences, access to care, physician behavior, and hospital policies [7,8,9,10,11,12,13,14]. Indications for CS vary in clinical and non-clinical settings, and differences are frequently observed depending on the ethnicity and region of the women [15]. To date, the real reason for the increased CS rates is not clear, and few studies have reported maternal or fetal risk profiles as the main indicators for the rise in CS rates [16].

To comprehend what may cause the increase in cesarean section rates and implement effective measurement methods and, furthermore, make suggestions for improvement, it has been documented that it is necessary to develop an international classification system for cesarean sections [8,15,16]. Many researchers published classification systems for cesarean sections that carried limitations in the interpretation of their results. In 2011, in a systematic review by Torloni and colleagues, 27 different classification systems for cesarean sections were studied, and it was concluded that the Robson classification system is the most suitable. The Robson tool uses the following six predefined obstetric characteristics of pregnant women: parity (nulliparous and multiparous women with and without previous cesarean sections), history of cesarean sections, mode of onset of labor (spontaneous, induced, or pre-labor cesarean section), number of fetuses (one or more than one), gestational age (preterm or term), and fetal presentation and lie (cephalic, breech, or transverse). Additionally, it classifies the pregnancies admitted for labor into one of ten classes (see Table 1) [17]. In 2015, the World Health Organization recommended the Robson classification as a universal standard to establish a joint control system in healthcare facilities [15].

The efficiency of the Robson classification system has been acclaimed over the last decade, and even more hospitals and countries are using it to monitor and evaluate their CS rates [18]. However, according to the authors’ knowledge, there are no available data after implementing the Robson classification in Greece. Data from relevant studies suggest that Greece is following the worldwide trend regarding high cesarean section rates during the last few decades [19,20,21], and it seems that Greece ranks first in Europe in cesarean sections [22]. As the Robson classification is categorized as a “women-based” classification model that essentially informs us who is going to undergo a CS, depending on maternal and pregnancy characteristics [16], it would be important for the multidimensional investigation of the increasing cesarean sections in Greece to be investigated simultaneously in the same sample based on the reasons and indications for CS.

The aim of this study was to implement the Robson classification system to identify trends in cesarean section rates in Greece and identify the groups of women who are the main contributors to the increasing rates. Moreover, the indicators for cesarean sections will be evaluated as per the Robson classification. We expect that the findings of this study will contribute to the design and focus of strategies aiming at making better use of cesarean sections in Greece.

## 2. Materials and Methods

This was a retrospective cross-sectional study conducted in a private hospital in Athens, Greece. This hospital conducts close to 10.000 deliveries annually. Moreover, it handles all types of pregnancies (including high-risk pregnancies) from all regions of Greece, and it includes a neonatal intensive care unit (NICU). The eligible participants consisted of women who gave birth between 1 January 2019 and 31 December 2019.

In 2019, according to the Hellenic Statistical Authority [23], 83,756 deliveries were carried out in Greece. At the study hospital, 8681 deliveries were performed. In the sample analysis, we included the medical records of 8572 women and their newborns. Women with gestational ages of ≥22 weeks and birth weights of ≥500 g were included. Women with stillborn fetuses/neonates (*n* = 73) were excluded from the sample due to a lack of data. A small percentage (*n* = 36) of women were not included in the study because there was no access to their medical records. The data were collected retrospectively by a trained data collector from the digital medical records of the women and their newborns. The retrieved data from the women’s medical records were as follows: age, smoking status during pregnancy, assisted reproduction technology, obstetric history (parity and previous CSs), onset of labor (spontaneous, induced, or CS before labor), fetal lie or presentation (cephalic, breech, transverse, or oblique), number of fetuses (single or multiple), gestational age (weeks), mode of birth (vaginal birth, operative vaginal birth, or CS), and indicator for CS. The data retrieved from the neonatal medical records were as follows: sex of the neonate (boy or girl) and birth weight.

For the data retrieval, collection, and analysis, an ethical approval was received by the hospital’s scientific board (1146/24-09-20). A signed consent form was not required by the women whose medical records were retrieved, as these women had already signed a GDPR form. The data concerning patient records and information were anonymous and de-identified before the analysis. The ratio of missing data was <3% for all variables in this study.

The variables were first tested for normality using the Kolmogorov–Smirnov test and a normal probability plot. Descriptive statistics of the study participants and variables were conducted. The results were presented as frequencies (*n*), percentages (%), and means ±SD. The data was extracted and analyzed using the IBM SPSS statistical software, version 21.0 (IBM Corporation, Somers, NY, USA). The analysis of the Robson classification data was calculated as described by the World Health Organization [24]. More specifically, in order to make the most of the information provided by the Robson classification, the data was reported in a standardized manner recommended by the World Health Organization (the “Robson Classification Reference Table”). The report table consisted of the following seven columns: column 1: group name and/or number and definition; column 2: total number of CSs in each group; column 3: total number of women delivered in each group; column 4: relative group size of each of the ten groups to the overall facility population (as a percentage); column 5: CS rate in each group (as a percentage); column 6: absolute group contribution of each of the ten groups to the overall CS rate (as a percentage); column 7: relative contribution of each of the ten groups to the overall CS rate (as a percentage).

## 3. Results

The data were collected from 8572 medical files of women and their deliveries. Their sociodemographic and obstetric characteristics are presented in Table 2. The mean age of the mothers was 34.16 years (SD = 4.90 years). The majority of women were Greek (94.6%) and nulliparous (57.6%). A minority of the women had smoked during their pregnancies (27.2%) and had used assisted reproductive technologies (13.3%). Additionally, 47.1% of the women gave birth between weeks 37^+0 days^ and 38^+6 days^. The pregnancies were singletons in 95.6% of the sample. During the study period, 60.9% of the sample had a cesarean section and 39.1% had a vaginal birth. From the total sample, 22.6% (1941/8572) had a previous cesarean section. Moreover, epidural/spinal anesthesia (or a combination of them) was performed in 91.4% of the cases. Regarding the newborns, 51.4% were boys, with the majority weighing 3000–3999 g and the percentage being 60.3%.

### 3.1. Robson TGCS

According to the study findings, as presented in Table 3, the women in Group 2 (nulliparous, singleton, cephalic, ≥37 weeks, and with induced labor or CS before labor), Group 5 (multiparous, singleton, cephalic, ≥37 weeks, and with previous CS), Group 4 (multiparous, singleton, cephalic, ≥37 weeks, and with induced labor or CS before labor), and Group 1 (nulliparous, singleton, cephalic, ≥37 weeks, and with spontaneous labor) accounted for the greatest contributions for all deliveries, with percentages of 34.5%, 19.8%, 12.0%, and 10.8%, respectively. Based on our study findings, it was found that the largest contributors to the overall cesarean section rate were Group 2 (34.6%), Group 5 (30.7%), Group 10 (11.7%), and Group 8 (7.0%). The above four groups were identified as “targeted groups”, contributing to almost 80% (4381/5224) of the total number of cesarean sections that were performed in the study hospital. The absolute group contribution for CS in Group 1 and Group 2 (25.25%) was substantially higher than Group 5 (18.66%), which indicates rising cesarean rates. For the nulliparous women, Group 1 contributed 6.9% to the overall CS rate, and 38.7% of the women within this group had CSs. Furthermore, 32.7% of the women in Group 2a underwent CSs. Additional subgroup analyses showed that 1247 out of the 2956 women in Group 2 and 29 out of the 1036 women in Group 4 had pre-labor CSs. Moreover, 93.3% of the women in Group 5.1 had CSs. Together, Group 6 and Group 7 contributed 7.3% to the overall CS rate, and the cesarean section rate was almost 100% within those groups.

### 3.2. Indications for Performing CS

The indications for CSs are listed in Table 4 and Figure 1. Cephalopelvic disproportions and previous cesarean sections were found to be the most common indications, with percentages of 41.7% and 34.6%, respectively, while the rest of the indicators gathered single-digit percentages. The indications per Robson group are shown in Table 3. Cephalopelvic disproportion was the leading indication in the majority of the following groups: Group 1 (86.9%), Group 2a (94.4%), Group 2b (84%), Group 3 (58.3%), Group 4a (90%), and Group 4b (79.3%). Additionally, a previous cesarean section was the most common indication in women with one or more previous CSs and a single cephalic term infant (Groups 5.1 and 5.2) at a rate of 96.6% and 95.3%, respectively. Breech or other malpresentations were the leading indications for CSs for Group 6 (96.8%), Group 7 (92.2%), and Group 9 (74.4%). Moreover, for Group 8, as expected, 96.2% had multiple gestations as an indication for CS. Additionally, for Group 10, the most common indications were placenta previa/placenta accreta (30.4%), previous CS (28.4%), and cephalopelvic disproportion (25.6%). A diagrammatic representation of the contributions of each indicator within the groups is displayed in Figure 2.

## 4. Discussion

According to our knowledge, this is the first study that represents birth data from Greece using the Robson classification. Our overall CS rate of 60.9% is higher than the corresponding rates in Europe (from 16.1% to 56.9%), as reported in a Euro-Peristat study [22], where Greece did not provide official data. It is noteworthy that in every Robson group, the cesarean section rates are much higher than the WHO Robson guideline suggests. The only exception is Group 9, where the recommendation and practice are 100% [24]. The Robson Groups 1, 2, and 5 tend to be the major contributors to the overall CS rate, contrary to studies from low-income settings [25], which may indicate that the CS rates will remain high in the future. In our study, the Robson Groups 2, 5, and 10 are the main contributors to the overall cesarean birth rate. Compared with the global reference for CS rates from the WHO MSC in Maternal and Newborn Health [6], which includes 42,637 women from 22 different countries, there are significant differences in the CS rates by Robson group. The cesarean section rates were higher for all Robson groups than the WHO MSC reference population, probably reflecting the trend of performing unnecessary CSs in Greece, especially in Groups 1, 2, 3, 4, and 10. In our study, the CS rate in Group 1 was 38.7%, while in the MSC reference population, it was under 10% (9,8%), and the WHO recommended a CS rate of <10% [24]. Though a CS rate of about 39.9% was reported to be achievable for Group 2 in the MSC reference population, the total CS rate was 61,0% in the Greek population, while the WHO CS rate is recommended to be about 20–35% in the latest published standards [22]. Group 1 + Group 2 accounted for 1/3 of the obstetric population, and similar results have been found in Brazil (33,3%) [26], Sri Lanka (38,1%) [27], France (38,2%) [28], and Canada (39,7%) [29]. In terms of multiparous women (excluding previous CSs), single cephalic, ≥37 weeks, and in spontaneous labor, induced labor, or CS before labor in Groups 3 + 4, the sum of the relative group contribution was 1.2%. The CS rates for Groups 3 and 4 were 3.2% and 4.7%, respectively. In the WHO MCS population, the CS rates for Groups 3 and 4 were 3.0% and 23.7%, respectively. For the above Robson groups, the WHO recommended a CS rate of <3% for Group 3 and around 15% for Group 4 [24]. For Group 5, rates of 50–60% are recommended, according to the WHO standards. However, in our study, Group 5 played a prevailing role, with a rate of >90%, and the relative group contribution for CS was four times higher than that of the MCS [6] reference population (30.7% vs. 7.2%). Our findings are consistent with the CS rates reported from Turkey [30], Cyprus [22], and Brazil [26]. The CS rates for Groups 6 and 7 were 99.3% and 96.2%, respectively, whereas those were reported to be 78.5% in Robson Group 6 and 73.8% in Group 7 for the MCS reference population. Moreover, the CS rate in Robson Group 8 (including those with CSs) was 96.8%, whereas for the MCS reference population, it was reported to be 57.7%, and rates of around 60% would be appropriate, according to the WHO [24]. Group 9 constituted 0.5% of the cases in the Greek population, with a CS rate of 100%. The CS in all transverse or oblique presentations was 88.6% in the WHO MSC population. Additionally, in Robson Group 10, while the CS was reported to be 25.3% for the MSC reference population and the WHO standards recommended a CS rate of around 30% [24], in our study, all singleton pregnancies of <37 weeks had a high cesarean section rate 79.1%. Our cesarean section rates are higher than those reported in other areas, such as the United Kingdom, Australia, Canada, New Zealand, Belgium [10], Ireland [31], USA [32], Sri Lanka [27], Palestine [33], and Austria [34].

In our study, the main indications for CSs were cephalopelvic disproportion and a previous cesarean section. The increased rates of cephalopelvic disproportion were the leading indications for Groups 1, 2a, 2b, 3, 4a, and 4b, explaining the percentages above 50% of CSs; however, the rates of overweight babies were about 3%. These rates (of CSs) are in contrast with other publications from different geographic regions, where the reported cephalopelvic disproportion rates range from 1% to 8% of CSs [35]. These findings suggest the need for further research on the overdiagnosis of cephalopelvic disproportion as an important factor in the increasing CS rates in the Greek population. Moreover, “the previous cesarean section” is the second most frequent indication for CS. In Robson Group 5, very low rates of VBAC (6.7% and 0.9% in Groups 5.1 and 5.2, respectively) were noted, although, according to other reports, a percentage of 60–72.8% of vaginal deliveries in women with previous CSs is achievable [36,37]. These findings highlight the need for developing practice guidelines for vaginal birth after cesarean section (VBAC) in Greece, which must take into consideration factors that, according to the latest systematic review, were associated with a successful vaginal birth after a cesarean section, such as maternal age, diabetes, obesity, hypertension, labor induction, previous vaginal birth, indications for the previous CS, Bishop score, and birth weight [38].

The results of this study offer new insight into the rising CS trend in Greece and explain the specific groups of women, according to the Robson classification, who are more likely to undergo this operation. Evidence-based strategies and interventions to reduce both primary and repeat CSs are needed to meaningfully reduce the overall cesarean birth rate. Careful evaluation of the inductions and grounds for elective CSs in nulliparous women and promoting TOLAC uptake could help achieve this.

### Strength and Limitations

There are many strengths to our study. Firstly, this was the first time that the Robson classification was implemented in Greece. Secondly, it should be emphasized that the sample was drawn from the biggest private hospital in Greece, where women from many Greek regions come to give birth to their children. Therefore, the sample of our study is considered representative of the Greek population in terms of size and social/financial and geographical characteristics. It is expected that our results will allow future comparisons between different hospitals in the same country and practice comparisons with other countries. For the assessment of data quality, the WHO recommends that the size of Robson Group 9 should be less than 1% and the CS rate in this group should be 100%. In our study, the size of Group 9 was 0.5% and the CS rate was 100%. Those results are in line with the WHO’s recommendation; therefore, it appears that the quality of the data was very good. However, our analysis was not without limitations. Some weaknesses of this analysis should be noted. The main limitation of our study was its retrospective nature. Additionally, it is important to emphasize that even though the sample size was large enough, these findings include only data from a private hospital and are not meant to be generalized to the whole country.

## 5. Conclusions

The prevalence of cesarean sections in Greece is very high compared to RTGCS and WHO data. These results emphasize the significance of implementing the Robson classification system as an evidence-based audit tool to determine the groups that are most associated with cesarean sections. This research revealed a high rate of cesarean sections, even in low-risk groups. Additional and more in-depth analyses will also be necessary for the Robson groups that contribute the most to the cesarean section rates. Educational interventions that engage women actively in planning for their births, such as childbirth preparation workshops from midwives, could increase women’s willingness to give birth vaginally (both for primiparous and multiparous women with a previous cesarean section). Moreover, the cesarean section rates above 90% in breech presentations and twin pregnancies and the very low rates of VBAC underline the need to strengthen the education of obstetricians and midwives on labor trials for the above women. As previous studies have found [39], differences between Greece, a country with high cesarean rates, and counties with low cesarean rates could be due, in part, to the increased number of deliveries in private hospitals and the absence of a midwifery-led maternity system. Our study is expected to assist policymakers in identifying effective interventions for the proper use of CSs in Greece and developing effective policies and protocols. These strategies could be financial interventions aimed at perinatal healthcare professionals and the development of different care-staffing models for labor units.

## Figures and Tables

**Figure 1 healthcare-11-00908-f001:**
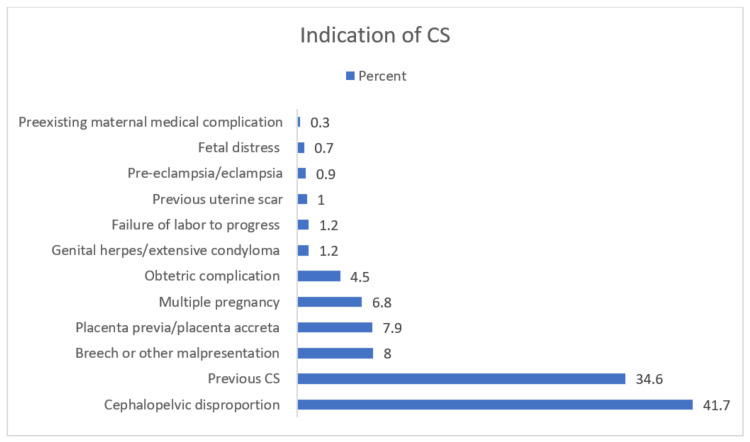
Indications for CSs.

**Figure 2 healthcare-11-00908-f002:**
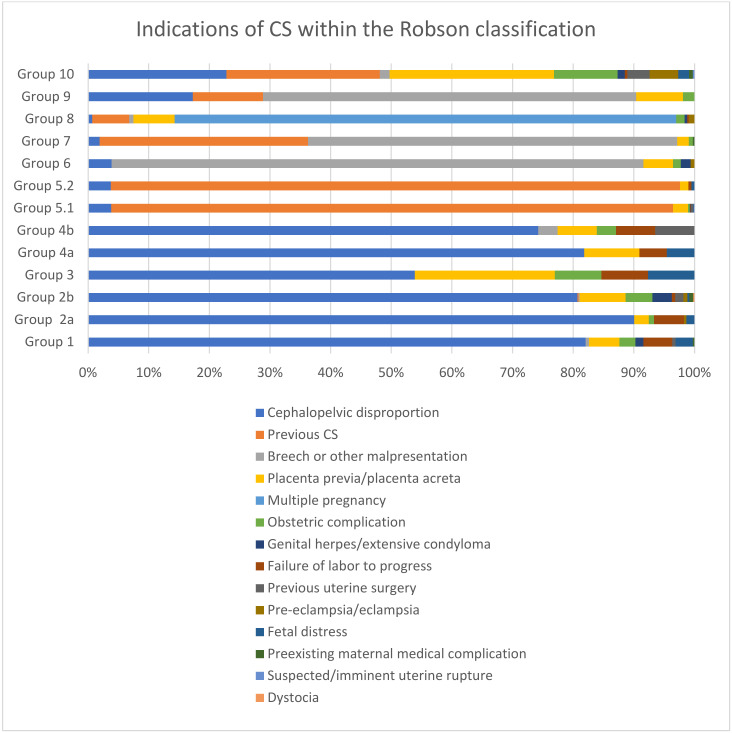
Indications for CSs within the Robson classification.

**Table 1 healthcare-11-00908-t001:** The Robson classification.

Robson Class	Description
1	Nulliparous women, single cephalic, ≥37 weeks, in spontaneous labor
2a	Nulliparous women, single cephalic, ≥37 weeks (labor induction)
2b	Nulliparous women, single cephalic, ≥37 weeks (cesareansection before labor)
3	Multiparous women (excluding previous CS), single cephalic, ≥37 weeks, in spontaneous labor
4a	Multiparous women (excluding previous CS), single cephalic, >37 weeks,(labor induction)
4b	Multiparous women (excluding previous CS), single cephalic, >37 weeks (cesarean section before labor)
5.1	One previous CS, single cephalic, ≥37 weeks
5.2	Two or more previous CSs, single cephalic, ≥37 weeks
6	All nulliparous women breeches
7	All multiparous women breeches (including previous CS)
8	All multiple pregnancies (including previous CS)
9	All abnormal lies (including previous CS)
10	All single cephalic, <37 weeks (including previous CS)

**Table 2 healthcare-11-00908-t002:** Sociodemographic and obstetric characteristics.

Characteristics		Frequency (*n*)	Percentage (%)
**Maternal age (years)**	mean (SD)	34.16 ± 4.90 (17–52)	
	<20	16	0.2
	20–29	1392	16.2
	30–39	6084	71.0
	≥40	1080	12.6
Νationality	Other	464	5.4
	Greek	8108	94.6
Parity	0	4938	57.6
	1	3030	35.3
	≥2	604	7.0
Previous CS	No	6631	77.4
	Yes	1941	22.6
Number of fetus	Single	8194	95.6
	Multiple	378	4.4
Gestational age (week)	<37^+0^	1160	13.5
	37^+0^–38^+6^	4039	47.1
	39^+0^–41^+6^	3367	39.3
	≥42^+0^	6	0.1
Mode of birth	Vaginal birth	3348	39.1
	Cesarean section	5224	60.9
Assisted reproductive technology	No	7430	86.7
	Yes	1142	13.3
Smoking status during gestation	No	6240	72.8
	Yes	2332	27.2
Type of anesthesia	None/Local	232	2.7
	Epidural/Spinal or combination of Epidural + Spinal	7845	91.4
	General anesthesia	435	5.1
	Combination of Epidural/Spinal + General anesthesia	60	0.7
Sex of baby	Male	4596	51.4
	Female	4351	48.6
Birth weight (g)	<2500	1055	11.8
	2500–2999	2234	25.0
	3000–3999	5397	60.3
	≥4000	261	2.9

**Table 3 healthcare-11-00908-t003:** The proportion of each Robson group, size of the group (%), CS rate (%), and their relative and absolute contributions to the overall CS rate (*).

Groups	Number of CS	Number of Women	Group Size ^1^ (%)	CS Rate ^2^ (%)	Absolute Group Contribution ^3^ (%)	Relative Group Contribution ^4^ (%)
1	359	928	10.8	38.7	4.19	6.9
2a	558	1709	20.0	32.7	6.51	10.7
2b	1247	1247	14.5	100.0	14.55	23.9
3	12	372	4.3	3.2	0.14	0.2
4a	20	1007	11.7	2.0	0.23	0.4
4b	29	29	0.3	100.0	0.34	0.6
5.1	1387	1487	17.3	93.3	16.18	26.6
5.2	213	215	2.5	99.1	2.48	4.1
6	278	280	3.3	99.3	3.24	5.3
7	102	106	1.2	96.2	1.19	2.0
8	367	379	4.4	96.8	4.28	7.0
9	43	43	0.5	100.0	0.50	0.8
10	609	770	9.0	79.1	7.10	11.7
Total	5224	8572	100.0	60.9	60.8	100.0

(*) Women with gestational ages of ≥22 weeks and birth weights of ≥500 g. Women with stillborn fetuses/neonates (*n* = 73) were excluded from the analysis. ^1^ Group size (%) = *n* of women in the group/total number of women delivered in the setting ×100. ^2^ Group CS rate (%) = *n* of CSs in the group/total number of women in the group ×100. ^3^ Absolute contribution (%) = *n* of CSs in the group/total number of women delivered in the setting ×100. ^4^ Relative contribution (%) = *n* of CSs in the group/total number of CSs in the setting ×100.

**Table 4 healthcare-11-00908-t004:** Indications for CSs within the Robson groups.

	Robson												
Indications for CS *	1 Ν(%)	2a Ν(%)	2b Ν(%)	3 Ν(%)	4a Ν(%)	4b Ν(%)	5.1 Ν(%)	5.2 Ν(%)	6 Ν(%)	7 Ν(%)	8 Ν(%)	9 Ν(%)	10 Ν(%)
Cephalopelvic disproportion	312 (86.9)	527 (94.4)	1074 (84)	7 (58.3)	18 (90)	23 (79.3)	55 (4.0)	8 (3.8)	12 (4.3)	3 (2.9)	3 (0.8)	9 (20.9)	156 (25.6)
Previous cesarean birth	0	0	3 (0.2)	0	0	0	1340 (96.6)	203 (95.3)	0	53 (52)	26 (7.11)	6 (14)	173 (28.4)
Breech or other malpresentation	2 (0.6)	0	3 (0.2)	0	0	1 (3.4)	0	0	269 (96.8)1	97 (92.2)	3 (0.8)	32 (74.4)	11 (1.8)
Placenta previa, placenta accreta	19 (5.3)	14 (2.5)	98 (7.9)	3 (25)	2 (10)	2 (6.9)	36 (2.6)	3 (1.4)	15 (5.4)	3 (2.9)	29 (7.9)	4 (9.3)	185 (30.4)
Multiple pregnancy	0	0	0	0	0	0	0	0	0	0	353 (96.2)	0	0
Obstetric complication (**)	10 (2.8)	5 (0.9)	57 (4.6)	1 (8.3)	0	1 (3.4)	4 (0.3)	0	4 (1.4)	1 (1.0)	6 (1.6)	1 (2.3)	72 (11.8)
Genital herpes/extensive condyloma	5 (1.4)	0	41 (3.3)	0	0	0	2 (0.1)	0	5 (1.8)	0	2 (0.5)	0	8 (1.3)
Failure of labor to progress	18 (5.0)	29 (5.2)	7 (0.6)	0	1 (5.0)	2 (6.9)	1 (0.1)	1 (0.5)	0	0	1 (0.3)	0	3 (0.5)
Previous uterine surgery (expect CS)	2 (0,6)	0	17 (1.4)	1 (8.3)	0	2 (6.9)	5 (0.4)	0	0	0	0	0	25 (4.1)
Pre-eclampsia/eclampsia	0	2 (0.4)	9 (0.7)	0	0	0	0	0	2 (0.7)	0	4 (1.1)	0	32 (5.3)
Fetal distress	11 (3.1)	8 (1.4)	4 (0.3)	0	1 (5.0)	0	1 (0.1)	1 (0.5)	0	0	0	0	12 (2.0)
Preexisting maternal medical complication	1 (0.3)	0	9 (0.7)	0	0	0	0	0	0	1 (0.4)	0	0	4 (0.7)
Suspected/imminent uterine rupture	0	0	0	0	0	0	1 (0.1)	0	0	0	0	0	2 (0.3)
Dystocia	0	0	2 (0.2)	0	0	0	0	0	0	0	0	0	0

(*) Women that may have had more than one indication for CS. (**) Including the following: antepartum hemorrhage, fetal macrosomia, intrauterine growth restriction, gestational diabetes, and/or other obstetric complications.

## Data Availability

Data sharing is not applicable.

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
