# Peer review of "Implementation of the Robson Classification in Greece: A Retrospective Cross-Sectional Study"

_healthcare, 2023, doi:10.3390/healthcare11060908_

Round 1

Reviewer 1 Report

The article analyses the use of Robertson classification for cesarean section indications in a Greek hospital.

The topic is important as it is ment to standardize the Indications for the intervention, and allow public health specialists and Obstetrics doctors to evaluate their activity.

Material and methods section is clear. It is a retrospective study, but the number of patients included and the selection criteria are somewhat defined. Maybe a little more details on the selection methods and the national register report could have been underlined.

The results are clearly depicted in tables and charts. 

The conclusion is also supported by the study. Future directions could be emphasized. 

Author Response

Thank you for your comments concerning our manuscript entitled “Implementation of Robson classification in Greece: a retrospective cross-sectional study”. Those comments are all valuable and very helpful for revising and improving our paper, as well as the important guiding significance to our research. We have studied these comments carefully and tried our best to revise and improve the manuscript. We sincerely hope that it will meet with approval.

Point 1: The article analyses the use of Robertson classification for cesarean section indications in a Greek hospital.The topic is important as it is ment to standardize the Indications for the intervention, and allow public health specialists and Obstetrics doctors to evaluate their activity. Material and methods section is clear. It is a retrospective study, but the number of patients included and the selection criteria are somewhat defined. Maybe a little more details on the selection methods and the national register report could have been underlined.

Response 1: Thanks very much for your valuable comment. We totally agree with your opinion and had added more details on the selection methods and the national register report. Specifically, we modified the second paragraph of the section: "materials and methods" to: ‘‘Ιn the year 2019 according to the Hellenic Statistical Authority, 83,756 deliveries were carried out in Greece. At the study Hospital, 8,681 deliveries were performed. In the sample analysis, we included 8572 women and their newborns medical records. Women with a gestational age ≥22 weeks gestation and birth weight ≥500 g were included. Women with stillborn fetuses/neonates (n=73) were excluded from the sample. A small percentage (n=36) of women were not included in the study because there was no access to their medical records…’’

Point 2: The results are clearly depicted in tables and charts. The conclusion is also supported by the study. Future directions could be emphasized. 

Response 2: Thank you for your comment. We have emphasized more future directions and modified the conclusion to: ‘‘The prevalence of cesarean section in Greece is very high compared to RTGCS and WHO data. These results emphasize the significance of implementing the Robson classification system in Greece as an evidenced-based audit tool to determine the groups most associated with cesarean section rates. This research revealed a high rate of cesarean sections even in low-risk groups. Educational interventions that engage women actively in planning for their birth such as childbirth preparation workshops from midwives could increase women’s willingness to give birth vaginally (both for primiparous and multiparous with a previous cesarean section). Moreover, cesarean section rates above 90% in breech presentation and twin pregnancies as well as very low rates of VBAC underline the need to strengthen the education of obstetricians and midwives on the trial of labor for the above women. Differences between Greece, a country with a high cesarean rate, and counties with a low cesarean rate could be partly due to the increased number of deliveries in private hospitals and the absence of a midwifery-led maternity system as previous studies have shown found [41]. Οur study is expected to assist policymakers to adopt multiple continuous interprofessional quality improvement interventions for the right use of CS in Greece and develop effective policies and protocols’’. Our study is expected to help policymakers to adopt multiple continuous interprofessional quality improvement interventions for the proper use of CS in Greece and to develop effective policies and protocols. These strategies could be financial interventions aimed at perinatal healthcare professionals and the development of different care staffing models for labor units’’.

Thanks again for your comments and suggestions for our paper.

We tried our best to improve the manuscript and made some changes in the manuscript according to your comments. We hope the corrections will meet with approval. Again, we appreciate your warm work. Looking forward to hearing from you.

Thank you and best regards.

Sincerely yours,

Paraskevi Giaxi

Reviewer 2 Report

Dear Authors,

The presented study tackles an issue of Implementation of Robson classification in Greece. I have read the article with a great interest. The study was conducted as retrospective cross-sectional study with reliably with appropriate selection of tests. Overall, I think that this article should be published, after minor revision.

Some issues require complementary information:

1.       The title should be in capital letters.

2.       Verse 20-22 Starting from “According to Robson classification………………..”I suggest changing it on more informative way- more elaborating the indications presented as a groups than the numbers.

3.       Verse 23-25 Starting from “Our study is expected to ….” What kind of effective strategies you mean?

4.       Introduction: I suggest elaborating the idea of Robson’s groups and the outcome in comparison with classical indications. Moreover, I suggest elaborating Robson classification eg. the difference between Robson 2a and 2b or 4a and 4b- it’s not clear for me.

5.       I suggest including in Discussion possible solutions for high incidence of CS in Greece.

Author Response

Thank you for your comments concerning our manuscript entitled “IMPLEMENTATION OF ROBSON CLASSIFICATION IN GREECE: A RETROSPECTIVE CROSS-SECTIONAL STUDY’’. Those comments are all valuable and very helpful for revising and improving our paper, as well as the important guiding significance to our research. We have studied these comments carefully and tried our best to revise and improve the manuscript. We sincerely hope that it will meet with approval.

Point 1: The title should be in capital letters.

Response 1: Thank you for the comment. We have made the required correction.

Point 2: Verse 20-22 Starting from “According to Robson classification………………..”I suggest changing it on more informative way- more elaborating the indications presented as a groups than the numbers.

Response 2: Thank you very much for your valuable comment. We totally agree with your opinion and have added a description for each group as follows: ‘‘According to Robson classification, in our study, the largest contributors to the overall rate of CS rate were: (a) nulliparous women with a single cephalic term pregnancy who either had labor induced or were delivered by cesarean section before labor- Group 2 (34,6%), (b) multiparous women with a single cephalic term pregnancy and at least one previous cesarean section- Group 5 (30,7%), (c) women with a single cephalic preterm pregnancy- Group 10 (11,7%) and (d)women with multiple pregnancies- Group 8 (7,0%)’’ .

Point 3: Verse 23-25 Starting from “Our study is expected to ….” What kind of effective strategies you mean?

Response 3: Thanks for your comment. We modified the sentence to make better sense: ‘‘Our study is expected to assist policymakers in Greece to plan further interventions for each subgroup of women to the overall CS rate to reduce the unnecessary CS rate’’. By strategies, we mean the interventions that could be developed in relation to the reduction of cesarean sections such as those developed in the conclusion: education, childbirth preparation workshops, financial interventions, and different staffing models for labor units.

Point 4: I suggest elaborating the idea of Robson’s groups and the outcome in comparison with classical indications. Moreover, I suggest elaborating Robson classification eg. the difference between Robson 2a and 2b or 4a and 4b- it’s not clear for me.

Response 4: Thanks for your warm work on our article. Regarding groups 2a, 2b, 4a, and 4b there was a mistake which we corrected in table 1 and now the differences between the groups are clearly visible. Also, as you suggest, we elaborate the idea of Robson classification and indications for cesarean section in the revised manuscript. Again, we appreciate your warm work earnestly, and thank you very much for your comments and suggestions for our paper.

Point 5: I suggest including in Discussion possible solutions for high incidence of CS in Greece.

Response 5: Thank you very much for your comment. We added in the conclusion possible solutions for the high incidence of CS in Greece as you and other reviewer mentioned.

We tried our best to improve the manuscript and made some changes in the manuscript according to your comments. We hope the corrections will meet with approval. Again, we appreciate your warm work. Looking forward to hearing from you.

Thank you and best regards.

Sincerely yours,

Paraskevi Giaxi

Reviewer 3 Report

The paper implemented the Robson Classification for the first time in Greece, which included the records of 8572 women giving birth in one private health facility. The study concluded cephalopelvic disproportion and previous cesarean section were the most common indication leading to cesarean section, which may useful to policymakers to identify effective strategies for specific subgroups of women to reduce the CS rate in Greece and improve outcomes. However, only discuss the specific subgroups of women who have higher cesarean section rates make this paper lack of novelty.

Author Response

Thank you for your comments concerning our manuscript entitled “Implementation of Robson classification in Greece: a retro-spective cross-sectional study”. Those comments are all valuable and very helpful for revising and improving our paper, as well as the important guiding significance to our researches. We have studied these comments carefully and tried our best to revise and imporve the manuscript. We sincerely hope that it will meet with approval.

Point 1: The paper implemented the Robson Classification for the first time in Greece, which included the records of 8572 women giving birth in one private health facility. The study concluded cephalopelvic disproportion and previous cesarean section were the most common indication leading to cesarean section, which may useful to policymakers to identify effective strategies for specific subgroups of women to reduce the CS rate in Greece and improve outcomes. However, only discuss the specific subgroups of women who have higher cesarean section rates make this paper lack of novelty.

Response 1: Thank you very much for your valuable comment. We would like to inform you that the Robson classification is applied for the first time in Greece. In addition, as we mention in the paper, in the latest Euro-Peristat study, Greece did not provide official data. In addition, for the first time results of the type of delivery and the indications for cesarean section are reported in such a large population sample in Greece. The above reasons we think make the article innovative. Finally, we added in the conclusion possible solutions for the high incidence of CS in Greece with the aim of improving the article.

Thank you and best regards.

Sincerely yours,

Paraskevi Giaxi